# Degradation of Lignin by Infrared Free Electron Laser

**DOI:** 10.3390/polym14122401

**Published:** 2022-06-14

**Authors:** Takayasu Kawasaki, Heishun Zen, Takeshi Sakai, Yoske Sumitomo, Kyoko Nogami, Ken Hayakawa, Toyonari Yaji, Toshiaki Ohta, Takashi Nagata, Yasushi Hayakawa

**Affiliations:** 1Accelerator Laboratory, High Energy Accelerator Research Organization, 1-1 Oho, Tsukuba 305-0801, Ibaraki, Japan; 2Institute of Advanced Energy, Kyoto University, Gokasho, Uji 611-0011, Kyoto, Japan; zen@iae.kyoto-u.ac.jp (H.Z.); nagata.takashi.6w@kyoto-u.ac.jp (T.N.); 3Laboratory for Electron Beam Research and Application (LEBRA), Institute of Quantum Science, Nihon University, 7-24-1 Narashinodai, Funabashi 274-8501, Chiba, Japan; sakai@lebra.nihon-u.ac.jp (T.S.); nogami@lebra.nihon-u.ac.jp (K.N.); hayakawa@lebra.nihon-u.ac.jp (K.H.); yahayak@lebra.nihon-u.ac.jp (Y.H.); 4Department of Physics, College of Science and Technology, Nihon University, 1-8-14 Kanda Surugadai, Chiyoda-ku 101-8308, Tokyo, Japan; sumitomo.yoske@nihon-u.ac.jp; 5SR Center, Research Organization of Science and Technology, Ritsumeikan University, 1-1-1 Noji-Higashi, Kusatsu 525-8577, Shiga, Japan; tyv27078@fc.ritsumei.ac.jp (T.Y.); ohta@fc.ritsumei.ac.jp (T.O.)

**Keywords:** infrared laser, lignin, depolymerization, vibrational excitation

## Abstract

Lignin monomers have attracted attention as functional materials for various industrial uses. However, it is challenging to obtain these monomers by degrading polymerized lignin due to the rigid ether linkage between the aromatic rings. Here, we propose a novel approach based on molecular vibrational excitation using infrared free electron laser (IR-FEL) for the degradation of lignin. The IR-FEL is an accelerator-based pico-second pulse laser, and commercially available powdered lignin was irradiated by the IR-FEL under atmospheric conditions. Synchrotron-radiation infrared microspectroscopy analysis showed that the absorption intensities at 1050 cm^−1^, 1140 cm^−1^, and 3400 cm^−1^ were largely decreased alongside decolorization. Electrospray ionization mass chromatography analysis showed that coumaryl alcohol was more abundant and a mass peak corresponding to hydrated coniferyl alcohol was detected after irradiation at 2.9 μm (νO-H) compared to the original lignin. Interestingly, a mass peak corresponding to vanillic acid appeared after irradiation at 7.1 μm (νC=C and νC-C), which was supported by our two-dimensional nuclear magnetic resonance spectroscopy analysis. Therefore, it seems that partial depolymerization of lignin can be induced by IR-FEL irradiation in a wavelength-dependent manner.

## 1. Introduction

The most abundant form of biomass on Earth is forests, and the development of efficient ways to use woody biomass is an urgent subject towards the establishment of a recycling system for environmental resources today. Lignocellulose is a main ingredient in wood and is composed of lignin, cellulose, hemicellulose, and acetylated polysaccharides [1]. Cellulose is a water-insoluble polymer held together by β1-4 glycoside bonds, and the biorefinery process of employing monomeric glucose derived from cellulose for bioethanol production is gathering attention worldwide as an alternative to fossil fuels [2,3,4]. In addition, research and development on cellulose fibers as advanced materials is currently thriving [5,6,7]. On the other hand, lignin, which makes up 15–35% of lignocellulose, is often removed and transferred to be burned as a fuel when cellulose is extracted from woody feedstock [8,9,10]. However, it has already been shown that monolignols and low-molecular weight lignin can be applied to the development of various functional materials such as UV absorbents, luminescent materials, anti-virus agents, and antioxidants [11,12,13,14]. Lignin-derived phenols can also be pro-oxidants for degrading pollutants [15]. In the conversion process from a lignin polymer to low-molecular weight phenol compounds, degrading the rigid β-O-4 ether (C-O-C) linkages that polymerize the aromatic molecules is an important step. As for the traditional methods for treating lignin, enzymes of microorganisms, UV-irradiation, and metal and acid/alkaline catalysis are frequently applied for the cleavage of the ether linkage after extracting the insoluble lignin from the woody feedstock using concentrated sulfuric acids with high temperature or microwave heating [16,17,18,19]. In particular, laccase from microorganisms is useful for not only oxidative degradation but also chemical modification to produce the valorized lignin. In fact, these enzymes can act under mild conditions and require no pollutants or chemicals. However, these approaches have both advantages and disadvantages in terms of processing time, running cost, versatility, practicality, and productivity. Towards the realization of a sustainable recycling system for woody biomass, it is further necessary to develop alternative methods for the treatment of rigid carbohydrate polymers.

We introduce an infrared free electron laser (IR-FEL) as a novel tool to obtain low-molecular weight aromatic compounds from polymerized lignin. The IR-FEL is an accelerator-based infrared pulse laser, and its oscillation manner is shown in Figure 1a [20,21,22]. An electron beam is emitted by a high radio frequency gun and is accelerated to close to the photon rate in a linear accelerator. The accelerated electron beam wiggles in the periodic magnetic field (so called undulator), and synchrotron radiation (SR) is produced along the tangent direction of the beam line. The IR-FEL is generated by the strong interaction of SR with the successive electron beam and is transmitted from a coupling hole in the center of a resonant mirror to a laboratory. A remarkable feature of the IR-FEL is that it gives intense vibrational excitation energy to the corresponding functional group’s mode-selectively, which can induce multi-photon absorption and dissociation reactions for various molecules in gas, liquid, and solid phases [23,24,25,26]. We have previously found that cellulose fibers can be dissociated, and glucose and low-molecular weight oligosaccharides can be recovered using an IR-FEL tuned to 9.1 μm (νC-O), 7.2 μm (δH-C-O), and 3.5 μm (νC-H) [27]. This study clearly showed that covalent linkages such as glycoside bonds can be dissected by IR-FEL irradiation in a wavelength-dependent manner at room temperature without any external heating. Therefore, it is expected that a similar protocol can be adopted to degrade lignin with wavelength selectivity. 

## 2. Materials and Methods

### 2.1. Materials

Lignin was purchased from NACALAI TESQUE, INC (Kyoto, Japan), and the characteristics from the company data sheet were as follows: the lignin was extracted from coniferous trees using sulfites, and chemical treatments including desulfonation, demethylation, and oxidation were performed during its production. The pH value of the product was adjusted to alkaline using sodium hydroxide. The product contains 8.0–14.0% methoxy groups and less than 15.0% water. Acetonitrile was purchased from FUJIFIM Wako Pure Chemical Co. (Tokyo, Japan). The water was purified using ultrapure water production equipment (PURELAB Flex UV, Veolia Jenets K.K., Tokyo, Japan). Dimethyl sulfoxide (DMSO)-D6 was purchased form NACALAI TESQUE, INC (Kyoto, Japan). 

### 2.2. Infrared Free Electron Laser Irradiation

An IR-FEL system at Kyoto University named KU-FEL was used for mid-infrared radiation [28], and the laser system at LEBRA of Nihon University was used for near-infrared radiation [29,30]. An electron beam is provided by an electron-linear accelerator, and the IR-FEL can be produced by the interaction between the synchrotron radiation stored in an optical resonator and the electron beam in the undulator. Then, the IR-FEL is extracted through a coupling hole with a diameter of about 1 mm for mid-IR FEL and 0.44 mm for near-IR FEL on one resonator mirror. The extracted IR-FEL beam is transported to user stations where irradiation experiments can be performed. The time structure of IR-FEL is composed of macro- and micro-pulses; one macro-pulse oscillates at a repetition rate of 1.4 Hz for mid-IR-FEL and 2.0 Hz for near-IR-FEL. The typical pulse duration of the former is 2 μs and that of the latter is 10 μs. One macro-pulse bunches several thousand micro-pulses, and the duration of each micro-pulse is less than 1 ps. The repetition rates of mid-IR FEL and near-IR FEL are both 2856 MHz in the full-bunch mode [31]. The FEL pulse energy integrated over the macro-pulse ranges from 1 to 20 mJ, and the beam diameter was set to about 300–400 μm above the sample surface using an off-axis parabolic mirror or a focusing lens (CaF_2_). 

As shown in the FT-IR spectrum of lignin used in this study (Figure 1b), there are two main bands in the near- and mid-infrared regions [32,33]. The near-infrared band from 3000 cm^−1^ (3.3 μm) to 3600 cm^−1^ (2.8 μm) contains a major O-H stretching mode (νO-H) and a minor C-H stretching mode (νC-H) as a shoulder band. The mid-infrared region at 1100–1600 cm^−1^ (6.3–9.1 μm) contains a C-H bending mode (δC-H), aliphatic C-C stretching (νC-C) and aromatic C=C stretching (νC=C) modes, and a C-O stretching mode (νC-O). The IR-FEL was tuned to 2.9 μm (3448 cm^−1^), 6.3 μm (1587 cm^−1^), and 7.1 μm (1408 cm^−1^), and these laser beams were introduced onto the lignin powder (about 5 mg) in a glass tube from the vertical direction for 10 to 15 min at room temperature (see Appendix A for the beam spectrum at each wavelength). The glass tube was sometimes shaken horizontally to give the radiation energy to the whole sample uniformly (see Appendix A for the experimental setup). After irradiation, the structural change of the lignin sample was analyzed as described below.

### 2.3. Synchrotron-Radiation Infrared Microspectroscopy (SR-IRM)

An SR-IRM observation was performed using the IR micro-spectroscopy beamline (BL-15) at the SR Center of Ritsumeikan University [34]. The beamline is equipped with a Nicolet 6700 FT-IR spectrometer and a Continuum XL IR microscope (Thermo Fisher Scientific, Tokyo, Japan). The lignin sample was suspended in water (10 mg/mL) and the mixture (10 mL) was placed on a stainless-steel base. After drying, measurements were performed in reflection mode with a 32 x Cassegrain lens at a 20 × 20 μm aperture. Spectra were recorded in the range of 900–4000 cm^−1^ at a resolution of 4 cm^−1^ with 32 scans. 

### 2.4. Electron-Spray Ionization Mass Spectroscopy

We employed JMS-T100CS AccuTOF CS (JEOL, Tokyo, Japan), and the measurement was performed at the ESI-positive mode. The liquid-chromatography conditions were as follows: ZORBAX Eclipse XDB-C18 (Agilent Technologies Japan, Ltd., Tokyo, Japan) was used for the column where the column size was 1 × 30 mm and the diameter was 3.5 μm. Elution was performed using A (water) and B (water/acetonitrile = 10/90(v/v)), and the gradient percentage was set to 1% of B. The flow rate was 0.05 mL/min, the column temperature was 25 °C, and the sample size was 10 μL. The lignin sample was suspended in water (10 mg/mL) and subjected to the mass analysis.

### 2.5. Nuclear Magnetic Resonance Spectroscopy 

The lignin sample (ca. 50 mg) was dissolved thoroughly in DMSO-d6 (600 μL) by ultrasound sonication. NMR spectra were recorded at 298 K on a Bruker AVANCE III HD 600 spectrometer (Bruker Corporation., Yokohama, Japan) equipped with a cryogenic probe and Z gradient (Bruker Biospin). The instrument was controlled using Bruker TopSpin 3.5. The two-dimensional (2D) ^13^C-^1^H heteronuclear single-quantum coherence (^13^C-^1^H HSQC) spectra were acquired using the Bruker pulse sequence ‘hsqcetgpsisp2.2’ with the following parameters: number of scans (NS), 128; spectral width (SWH), 7212 Hz (^1^H) and 22645 Hz (^13^C); acquisition time (AQ), 142 ms (^1^H) and 11 ms (^13^C). The ^13^C-^1^H 2D heteronuclear multiple bond correlation (HMBC) spectra were acquired using the Bruker pulse sequence “hmbcetgpl3nd” with the following parameters: number of scans (NS), 128; spectral width (SWH), 7212 Hz (^1^H) and 33213 Hz (^13^C); acquisition time (AQ), 284 ms (^1^H) and 7.7 ms (^13^C). All the chemical shifts were referenced to the DMSO-d6 solvent peak (δC = 39.5 ppm and δH = 2.49 ppm). NMR data were processed and analyzed with TopSpin 3.5 and Sparky [35], respectively. 

## 3. Results

### 3.1. Irradiation Effect of IR-FEL at Near-Infrared Wavelength

First, we show an irradiation effect of the IR-FEL tuned to 2.9 μm (Figure 2). In the optical microscopy image (Figure 2a, right panel), there are several cracks on the surface of dry lignin, the size of the crack was smaller, and the number was more greatly increased after irradiation than before irradiation. The black color of the surface was slightly decolorized to brown. These morphological changes indicate that the infrared laser irradiation caused some physical damage to the structure of the lignin. In the SR-IRM spectra (Figure 2a, left and middle panels), the broad band around 3400 cm^−1^ (νO-H) was substantially decreased (left panel) and two strong peaks at about 1140 cm^−1^ (aromatic C-H deformation vibration) and 1050 cm^−1^ (C-O stretch vibration) were decreased (middle panel) after the irradiation (red) compared to the original lignin (black). In addition, the half width of the strong peak at about 1590 cm^−1^ (which covers aromatic C=C and aliphatic C-C stretch vibrations) was slightly shortened by the irradiation. Therefore, it seems that some large conformational changes can be induced by vibrational excitation at about 3 μm.

To know how the conformation of lignin was changed by the IR-FEL irradiation in detail, we measured ESI-MS (Figure 2b). In the spectrum of lignin before irradiation (upper, #1), the main peak is observed at 164 Da in the low-molecular weight region (100–200 Da) and this peak disappeared after irradiation (#2). Instead, mass peaks at 198, 173, 158, 141, and 128 Da were mainly detected. If 173 Da is a sodium ion adduct of coumaryl alcohol (150 Da), its fragment peaks (128, 141, and 158 Da) can be assigned as follows: 158 = 173–15(CH3); 141 = 158–17(OH); and 128 = 141–13(CH). The peak at 198 Da may be a hydrated adduct of coniferyl alcohol (180 Da). The mass chromatogram of the peak at 173 Da (below) showed that the fraction was eluted after 4.5 min in the column, and its quantity was higher in the sample after the irradiation (red) than that of the non-irradiation sample (black). Therefore, it was suggested that coumaryl alcohol was more abundant after the IR-FEL irradiation than in the original lignin. 

### 3.2. Irradiation Effect of IR-FEL at Mid-Infrared Wavelengths

Second, we looked at the irradiation effects of IR-FEL tuned to 6.3 μm and 7.1 μm (Figure 3). In the optical microscopy image (Figure 3a, right), the number of cracks increased and the color of the surface of lignin was changed from black to reddish brown after these irradiations (upper and middle) compared to before irradiation (bottom). In particular, the morphological change caused by irradiation at 7.1 μm (upper) was more remarkable than with irradiation at 6.3 μm (middle). SR-IRM analysis (left) showed that a strong band around 1590 cm^−1^ was largely decreased, accompanied by an increase in the shoulder band at a higher wavenumber, and a middle peak at 1130–1145 cm^−1^ was slightly decreased after 7.1 μm irradiation (blue) compared to the original lignin (black). The shoulder band around 1635 cm^−1^ contains C=O stretching vibrational mode, and the appearance of this shoulder is quite different from the effect of 2.9 μm irradiation (Figure 2a, red). Interestingly, the spectral pattern was not changed much in the case of irradiation at 6.3 μm (green), which may imply that the structural alteration was minor compared to that in the case of 7.1 μm. Thus, we investigated the effect of irradiation at 7.1 μm in detail as described below. 

In the ESI-MS spectra in the high-molecular weight region (Figure 3b), multivalent ions were detected before and after irradiation, and the interval value was increased from 74 before irradiation to 136 after irradiation where the difference was 62, while the mass peak at 838 Da was not changed. The difference value of the interval of multivalent ions can be expanded to 2 x (16 + 15), which corresponds to two oxygens and two methyl groups, or 2 x (17 + 14), which corresponds to two hydroxy and two methylene groups. In addition, several fragment ions were apparently observed in the low-molecular weight region below 200 Da after irradiation compared to the sample before irradiation (brown dotted enclosure). 

Next, we analyzed the low-molecular weight region below 200 Da in detail (Figure 3c). A mass peak at 164 Da was mainly detected before irradiation (black), and several fragment peaks including 158, 173, 185, and 191 Da were observed after irradiation (blue); 173 may correspond to a sodium ion adduct of coumaryl alcohol (150 Da), 158 is its fragment peak (-15[CH3]), 185 is considered to be a fragment peak from hydrated coniferyl alcohol (198-13[CH]), and 191 could possibly be a sodium ion adduct of vanillic acid (168 Da). In the mass chromatography analysis (below), the volume of the fraction containing these fragment ions after irradiation (#2) was larger and eluted more belatedly than the fraction before irradiation (#1). These results indicate that infrared laser irradiation at 7.1 μm induced remarkable fragmentation of the main chain of lignin, and it is interesting that vanillic acid appeared after irradiation.

To further evaluate the production of vanillic acid, we carried out an NMR analysis (Figure 4a–c). Prior to the irradiation experiment, we measured a 2D ^13^C-^1^H HSQC NMR spectrum of this lignin sample (Figure 4a), and the spectrum exhibited the presence of only a small amount of syringyl (S) structural unit, as the intensities of the signals for C_2,6_/H_2,6_ of the S structural unit were weak (see the region “S_2,6_”), while strong signals for C_2_/H_2_, C_5_/H_5_, and C_6_/H_6_ of guaiacyl (G) structural unit were observed (see the regions “G_2_” and “G_5,6_”). This indicates the presence of a larger amount of the G structural unit over the S structural unit. The desorption of two methoxy groups on the aromatic ring (in the case of syringyl group) by the irradiation is unlikely, and thus the reduction in the number of multivalent ions in the high-molecular weight region and the fragmentation in the low-molecular weight region in the ESI-MS analysis (Figure 3b) may imply that the aliphatic chain crosslink containing hydroxy groups was cleaved by the irradiation. 

The 2D ^13^C-^1^H HSQC and 2D ^13^C-^1^H HMBC spectra of the sample after irradiation at 7.1 μm are superimposed in Figure 4c. The signals of the 2D ^13^C-^1^H HSQC spectrum (red) at (δC/δH) = (113.1/7.45), (114.2/6.70), (122.6/7.35), and (55.5/3.74) turned out to belong to the C_2_/H_2_, C_5_/H_5_, C_6_/H_6_, and C_MeO_/H_MeO_ of the vanillic acid unit, respectively. Additionally, the signals of the 2D ^13^C-^1^H HMBC spectrum (blue) at (δC/δH) = (113.0/7.35), (146.5/7.44), (146.5/3.74), (148.4/7.44), (148.4/6.70), (122.7/7.45), (169.7/7.35), and (169.7/7.44) were found to be the correlation signals for the C_2_/H_6_, C_3_/H_2_, C_3_/H_MeO_, C_4_/H_2_, C_4_/H_5_, C_6_/H_2_, Cα/H_6_, and Cα/H_2_ of the vanillic acid unit, respectively. These chemical shift values were similar to, if not the same as, those reported previously [36]. We noticed that there are signals that probably belong to the C_2_/H_2_, C_5_/H_5_, and C_6_/H_6_ of the vanillic acid unit in the 2D ^13^C-^1^H HSQC spectrum of the sample before irradiation (Figure 4b, red). However, the intensities of these signals were weak, indicating that the amount of the vanillic acid unit in the sample before irradiation was low. Since the intensities of the signals of the vanillic acid unit in the sample after the irradiation were strong, the amount of the vanillic acid unit seems to be increased by IR-FEL irradiation.

## 4. Discussion

This study demonstrates for the first time that near-IR and the mid-IR FELs can induce the partial fragmentation of lignin through vibrational excitation targeting the monolignol units to release several aromatic compounds (Figure 5). In the case of 2.9 μm, vibrational excitation energy can be deposited into the aliphatic chain containing OH groups, which can lead to the release of coumaryl alcohol and coniferyl alcohol derivatives. In the case of 7.1 μm, the vibrational excitation energy can be broadly absorbed in the aromatic νC=C and aliphatic νC-C of the main chains in the polymerized conformation, resulting in the cleavage of the ether linkages. Regarding the reaction mechanism, it can be a reasonable interpretation that multiphoton absorption generates vibrationally hot molecules that can produce thermochemically stable dissociation products in a wavelength-dependent manner, as shown in the gas phase reactions such as the isomerization of 2,3-dihydrofuran to cyclopropanecarboxaldehyde and cis/trans-crotonaldehyde under the IR-FEL irradiation [37]. As for the solid-phase molecules, Zavalin et al. reported that the IR-FEL was tested to ablate the biological tissue, and the collagen protein was degraded by the irradiation targeting C-N bond where the carbocation and the nitrogen radicals were generated under irradiation conditions [38]. In our previous study on cellulose degradation, we found that glycoside bonds could be cleaved by specific irradiation at the resonant wavelengths of 9.1 μm (νC-O), 7.2 μm (δH-C-O), and 3.5 μm (νC-H) to release glucose together with sugar oligomers [27]. Additionally, melanin pigment was structurally modified by the IR-FEL tuned to 5.8 μm (νC=O of carboxylate) to release pyrrole rings [39]. Based on those examples, it can be estimated that the vibrational excitation against the monolignol unit in the polymerized lignin induces radical cleavage of the crosslink between aromatic rings and dissociation of the aggregate structure to produce several lignin monomers. As shown in these studies, it is quite interesting that the products are varied dependent on the difference of the excitation wavelengths resonating to the individual functional groups. Nonetheless, we tested only three wavelengths of irradiation on lignin in this study, and there are many other absorption bands, for example, at about 1500 cm^−1^ (6.7 μm) and at 1000–1200 cm^−1^ (8.3–10 μm) in the infrared absorption spectrum of the lignin (Figure 1b). It is likely that many other aromatic compounds can be obtained if the lignin is vibrationally excited at these wavelengths. Such a study on the screening of irradiation wavelengths for obtaining various types of aromatic molecules should be an important subject for the application of IR-FEL to the treatment of lignin polymers.

Together with the results on cellulose and lignin, the IR-FEL irradiation technique is promising for the degradation of woody carbohydrates to produce constituent monomers by tuning the wavelengths to resonate with functional groups such as C-O, C-C, C=C, and O-H bonds. This treatment would contribute to the industrial application of phenolic compounds. For example, vanillic acid has pharmacological characteristics, in addition to being reduced to vanillin, which is an important input for the food and cosmetics industry [40,41]. Although the present study is a preliminary result showing the physicochemical depolymerization of the lignin using an infrared laser, two important aspects can be pointed out as follows: one is that the laser-induced dissociation reaction can proceed within several micro seconds because the macro-pulse duration of the IR-FEL is about 2–10 μs; the other is that the vibrational excitation reaction requires no organic solvents and no high temperature and pressure. These technical features offer advantages over conventional systems that use microwaves, microbial enzymes, and organic metals for the degradation of carbohydrate polymers in terms of the rapidness of the treatment, negative emissions, and biohazard avoidance. Nonetheless, a concern is the productive efficiency of the lignin monomers after laser irradiation. One batch system includes only about 5 to 10 mg of original lignin sample as a starting material. Based on the total ion chromatograms analysis, the yield of a fraction containing vanillic acid can be estimated to be less than 1% of the total fraction. Therefore, the sample volume that can be irradiated by the laser at once is limited and it is necessary to perform the irradiation process continuously to perform the large-scale production of lignin monomers. In addition, an automatic sample changer can be introduced on the beam line under the irradiation conditions for the treatment of a large number of samples. Accordingly, we plan to optimize the irradiation parameters such as pulse energy, pulse frequency, irradiation time, and irradiation wavelengths, for the purposes of the improvement in the productive efficiency of the lignin monomers, in future studies.

## 5. Conclusions

Infrared free electron laser (IR-FEL) is an accelerator-based, high-power pulse laser. In this study, the IR-FEL was tested to depolymerize lignin. The commercially available lignin was irradiated by the IR-FEL under atmospheric conditions, and the SR-IRM analysis showed that the band intensities at 1050 cm^−1^, 1140 cm^−1^, and 3400 cm^−1^ were decreased and that the half width of the strong peak at about 1590 cm^−1^ was slightly shortened after irradiation at about 3.0 μm compared to the original lignin. In the case of irradiation at 7.1 μm, a strong band at around 1590 cm^−1^ was largely decreased, accompanied by an increase in the shoulder band at higher wavenumbers and the middle peak at 1130~45 cm^−1^ being slightly decreased. The irradiated samples were decolorized from black to reddish brown. These morphological changes indicate that the strong vibrational excitation targeting the core monolignol unit can induce large conformational changes and perturbations in the ether bonds in the lignin. In addition, ESI-MS analysis followed by 2D NMR analysis showed that the covalent bonds in the aliphatic chains were cleaved and several phenolic compounds such as coumaryl alcohol and hydrated coniferyl alcohol were produced by those irradiations, and vanillic acid was obtained after irradiation at 7.1 μm. This study suggests that strong vibrational excitation energy can trigger the dissociation of a part of the polymerized lignin, which can release several low-molecular weight aromatic compounds. In future, various infrared wavelengths should be tested to dissociate the insoluble lignin from woody feedstock, and the irradiation parameters of the IR-FEL should be optimized to produce functional lignin monomers that can contribute to the development of sustainable materials for industrial applications. 

## Figures and Tables

**Figure 1 polymers-14-02401-f001:**
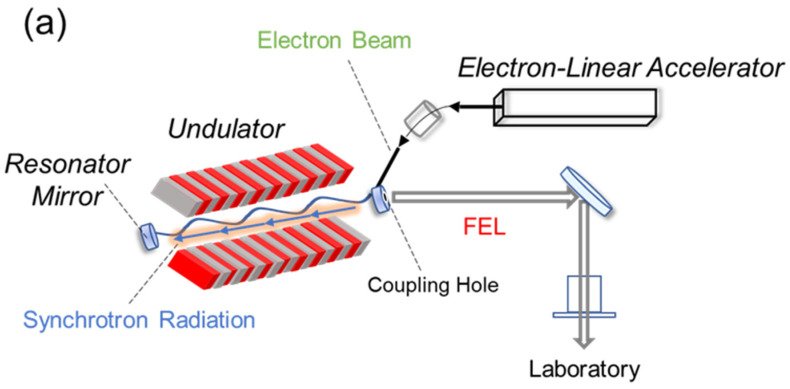
(**a**) Exterior of IR-FEL oscillation system. The instrument is constructed mainly of three devices (*italic*) in a synchrotron radiation facility: electron-linear accelerator, undulator (periodic magnetic field), and resonator mirrors. The laser beam is extracted from a coupling hole (0.4–1.0 mm in diameter) and transported to a laboratory. (**b**) FT-IR spectrum of lignin at mid-infrared (900–2000 cm^−1^) and near-infrared (2500–3900 cm^−1^) regions. Each region contains several molecular vibrational modes (underlines). Red arrows indicate absorption bands (3448, 1587, and 1408 cm^−1^) targeted for IR-FEL irradiation.

**Figure 2 polymers-14-02401-f002:**
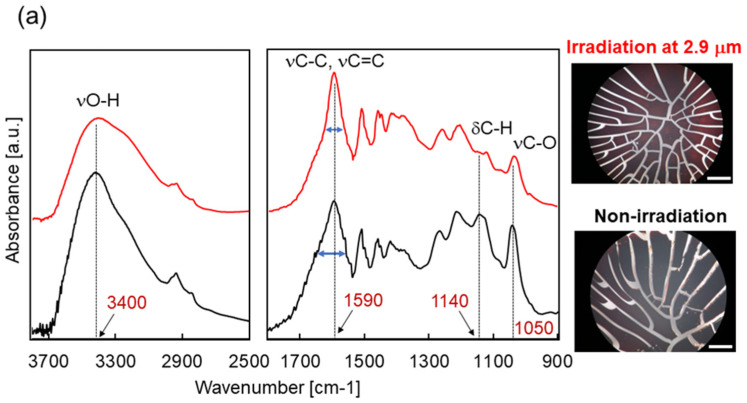
Irradiation effect of IR-FEL at 2.9 μm. (**a**) Infrared microspectroscopy observation of lignin before irradiation (black) and after irradiation (red) at near-infrared (left) and mid-infrared (middle) regions. The right panels show optical microscope images for surfaces of the dry lignin before and after irradiation. White bar: 200 μm. (**b**) ESI-MS chromatography analysis. Upper: mass spectra at low-molecular weight region; #1: lignin before irradiation; #2: lignin after irradiation. Below: mass chromatogram of a peak at 173 Da. Black: sample before irradiation; red: sample after irradiation.

**Figure 3 polymers-14-02401-f003:**
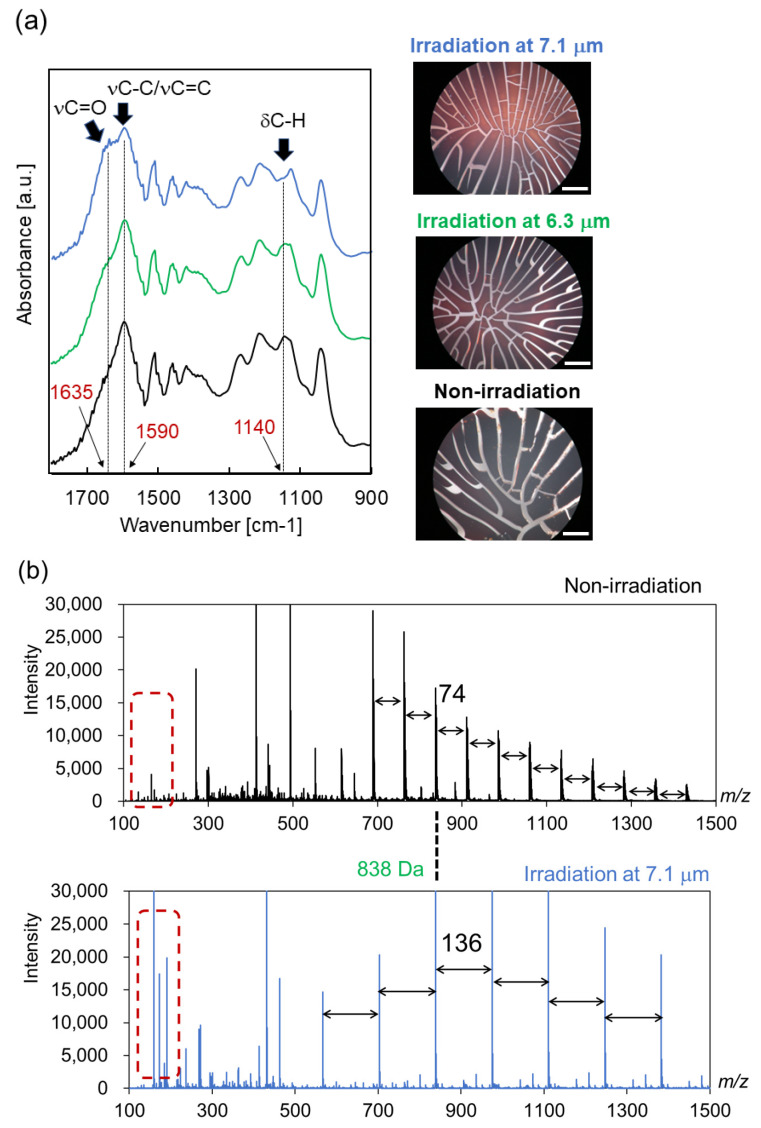
Irradiation effect of IR-FEL at mid-infrared wavelengths. (**a**) Infrared microspectroscopy observation of lignin before (black) and after irradiation at 6.3 μm (green) and 7.1 μm (blue). The right photographs show optical microscope images for surfaces of the dry lignin before and after irradiation. White bar: 200 μm. (**b**) ESI-MS spectra in the high-molecular weight region. Upper: lignin before irradiation; below: lignin after irradiation at 7.1 μm. (**c**) ESI-MS chromatography analysis in the low-molecular weight region. Black: lignin before irradiation; blue: lignin after irradiation at 7.1 μm. Upper: mass spectra; below: mass chromatograms of lignin before (#1) and after (#2) irradiation.

**Figure 4 polymers-14-02401-f004:**
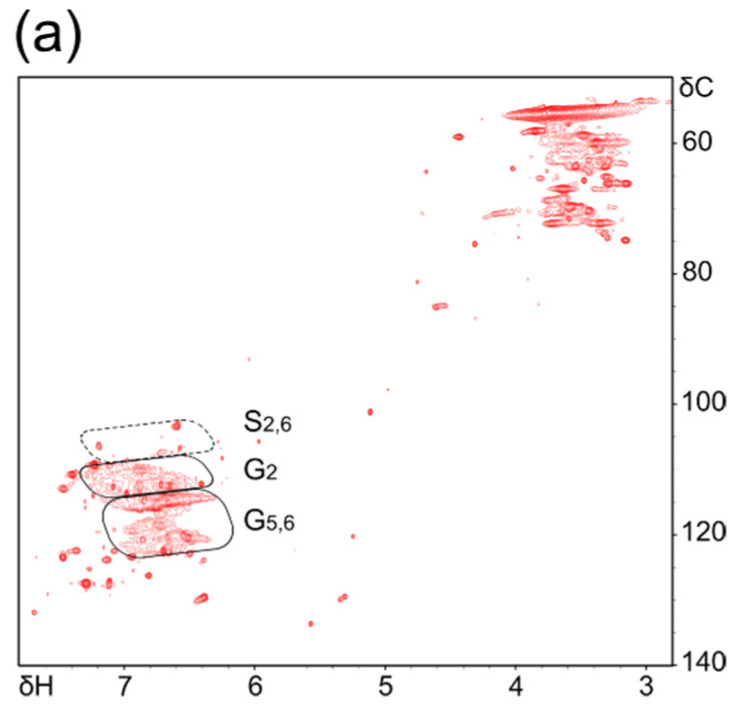
2D NMR analysis. (**a**) Superimposition of partial 2D ^13^C-^1^H HSQC of lignin sample before irradiation in DMSO-d6. Vertical and horizontal axes are chemical shifts in parts per million (ppm) for ^13^C (δC) and ^1^H (δH), respectively. Typical aromatic spectral regions for ^13^C and ^1^H of syringyl (S) and guaiacyl (G) moieties are indicated. (**b**,**c**) Superimposition of partial 2D ^13^C-^1^H HSQC (red) and 2D ^13^C-^1^H HMBC (blue) spectra of lignin before (**b**) and after (**c**) irradiation in DMSO-d6. The signals corresponding to vanillic acid are labeled (refer to upper right structural formula). Vertical and horizontal axes are chemical shifts in parts per million (ppm) for ^13^C (δC) and ^1^H (δH), respectively.

**Figure 5 polymers-14-02401-f005:**
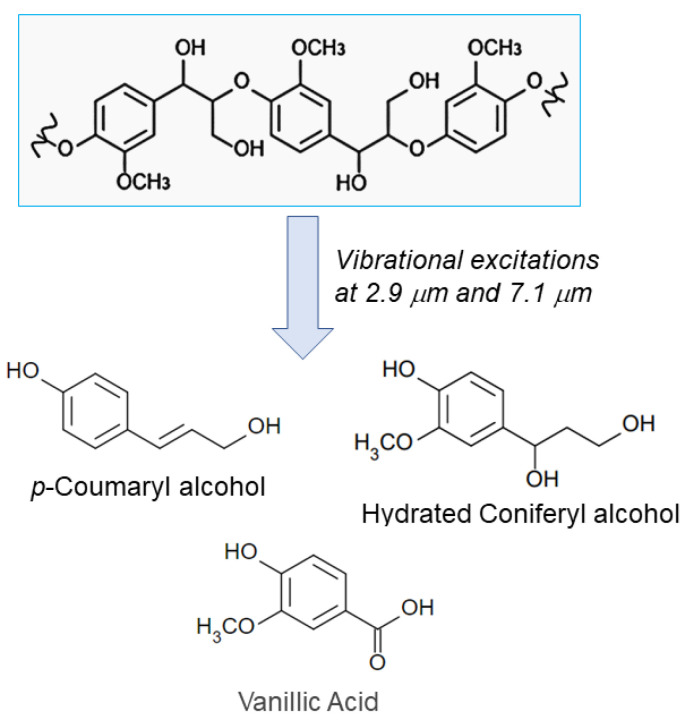
Degradation of lignin by vibrational excitation at 2.9 μm and 7.1 μm. Three lignin monomers were identified by the ESI-MS and 2D NMR analyses after the IR-FEL irradiations.

## Data Availability

The original data are available on request from the corresponding author.

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
