# Peer review of "Degradation of Lignin by Infrared Free Electron Laser"

_polymers, 2022, doi:10.3390/polym14122401_

Round 1

Reviewer 1 Report

The article “Degradation of Lignin by Infrared Free Electron Laser” presents results about obtaining lignin derived-phenols. However, some corrections are necessary:

Abstract

- Rephrase the last sentence;

Introduction

- Line 40: and Hemicelluloses? These other polysaccharides must be mentioned;

- Line 49: lignin-derived phenols can be also pro-oxidants in degrading pollutants (Effect of Lignin-Derived Methoxyphenols in Dye Decolorization by Fenton Systems, https://doi.org/10.1007/s11270-015-2703-0);

Experimental

- Create a topic about reagents;

- Line 110: correct to “diameter”;

- It is necessary to mention the origin of this commercial lignin, in order to better understand the treatment products. For example, grasses often have HGS lignin;

- Line 143: Were the authors able to solubilize lignin in pure water?

Discussion

- It is necessary to discuss not only the effect of the treatment, but also the possible application of the phenols obtained. For example, vanillic acid has pharmacological characteristics, in addition to being reduced to vanillin, which is an important input for the food and cosmetics industry.

Author Response

Dear Reviewer 1

Thank you very much for reviewing our paper. We made responses to all of your comments as described below and revised the manuscript. We would appreciate it if you could review the responses and the revised paper.

Comment 1: Rephrase the last sentence;

Answer 1: Thank you for the comment. We guess that you may indicate the last sentence may be overstatement and almost the same as the former phrase “molecular vibrational excitation by using infrared free electron laser (IR-FEL) for the degradation of lignin” in line21-22. Therefore, we changed the last sentence as follows:

“it seems that partial depolymerization of lignin can be induced by IR-FEL irradiation in a wavelength-dependent manner.”

We would appreciate it if you could consider this revision.

Comment 2: Line 40: and Hemicelluloses? These other polysaccharides must be mentioned;

Answer 2: Thank you for your suggestion. That is absolutely correct. We added hemicellulose and acetylated polysaccharides to this part (line 38-39).

Comment 3: Line 49: lignin-derived phenols can be also pro-oxidants in degrading pollutants (Effect of Lignin-Derived Methoxyphenols in Dye Decolorization by Fenton Systems, https://doi.org/10.1007/s11270-015-2703-0);

Answer 3: Thank you so much for the valuable suggestion and for giving us the reference. We described the lignin-derived phenols in this part (line 48-49) and added the reference [15] as you suggested.

Comment 4: Create a topic about reagents;

Answer 4: Thank you for the comment on the materials. We added the description about the reagents used in this study in the section of 2.1 Materials (94-103).

Comment 5: Line 110: correct to “diameter”;

Answer 5: Thank you so much for indicating the typographical mistake. We corrected the word as you suggested (line111).

Comment 6: It is necessary to mention the origin of this commercial lignin, in order to better understand the treatment products. For example, grasses often have HGS lignin;

Answer 6: Thank you so much for the valuable suggestion. We asked for the company and got the detail about the product. We added a sentence about the origin of the lignin used in this study to the 2.1 Materials section as follows (line 95-100):

Lignin was purchased from NACALAI TESQUE, INC (Kyoto, Japan), and the characteristics from the company data sheet were as follows: the lignin was extracted from coniferous trees by using sulfites, and chemical treatments including desulfonation, demethylation, and oxidation were performed for the production. The pH value of the product was adjusted to alkaline by using sodium hydroxide. The product contains 8.0-14.0% of methoxy groups and less than 15.0% of water.

Comment 7: Line 143: Were the authors able to solubilize lignin in pure water?

Answer 7: Thank you for the question. We used distilled water to handle the lignin sample and it seems to be dissolved visually. However, there is a possibility that the lignin is not completely solubilized in water at microscopic level. Therefore, we replaced the word “dissolved” to “suspended” (line 152). Thank you for the suggestion.

Discussion

Comment 8: It is necessary to discuss not only the effect of the treatment, but also the possible application of the phenols obtained. For example, vanillic acid has pharmacological characteristics, in addition to being reduced to vanillin, which is an important input for the food and cosmetics industry.

Answer 8: Thank you for the nice comment. Exactly, the application of the product after the laser treatment of lignin is very important. We added the sentences and new references [40,41] about the application of the phenol derivatives to the industrial fields after the laser treatment (line 355-358) as you suggested.

That’s all.

Sincerely yours

Takayasu Kawasaki, Ph.D.

Accelerator Laboratory, High Energy Accelerator Research Organization, 1-1 Oho, Tsukuba, Ibaraki 305-0801, Japan.

Phone: +81-29-864-5200-2014, Fax: +81-29-864-3182

Reviewer 2 Report

This paper concerns the interaction of infrared free electron laser with lignin to degrade them and more precisely to depolymerize them.

This work is pretty novelty and bring some new and interesting information.

However some modification must be done before publication.

- The first is about the form and especially on the Figure 3. It is really a Figure but a compilation of 7 Figures. Therefore the legend represents 16 lines. It would be better to separate them or to move some part in supporting information.

- The others is about the substrate of this articles.

In Introduction (line 47) change “recently discovered” by already shown”. The role of lignin and monolignol described below is well known and widely exploited.

The paragraph from line 82 to line 92  must be move in experimental part

-It is mandatory to give all the characteristics of the lignin used in this work. “alkaline treatment lignin is not enough. (lignin from coniferous for example, what kind of alkaline treatment, the purity of the lignin, the molar masses and so on..) 

-The yield of the transformation must be indicated. (quantum yield, chemical yield and so on..

- Is the sample store in the same conditions (humidity, temperature) before irradiation and analyses.

-It is very difficult to appreciate the changes on IR microspectroscopy observations and consequently the chemical modifications.

Author Response

Dear Reviewer 2

Thank you very much for reviewing our paper. We made responses to all of your comments as described below and revised the manuscript. We would appreciate it if you could review the responses and the revised paper.

Comment 1: The first is about the form and especially on the Figure 3. It is really a Figure but a compilation of 7 Figures. Therefore the legend represents 16 lines. It would be better to separate them or to move some part in supporting information.

Answer 1: Thank you for the valuable suggestion. We separated the NMR data from the other analytical data and put together in new Figure 4 (lines 244-270, 305-312). New Figure 3 contains only SR-IRM and ESI-MS data (lines 208-242, 280-288).

Comment 2: The others is about the substrate of this articles. In Introduction (line 47) change “recently discovered” by already shown”. The role of lignin and monolignol described below is well known and widely exploited.

Answer 2: Thank you very much for the suggestion. That is absolutely correct. We changed the word as you suggested (line 45).

Comment 3: The paragraph from line 82 to line 92  must be move in experimental part

Answer 3: Thank you for the suggestion. We moved this paragraph to the experimental part as you indicated (line 122-134).

Comment 4: It is mandatory to give all the characteristics of the lignin used in this work. “alkaline treatment lignin is not enough. (lignin from coniferous for example, what kind of alkaline treatment, the purity of the lignin, the molar masses and so on..) 

Answer 4: Thank you for the suggestion. We asked for the company and got the detail about the lignin used in this study and added a description about its characteristics as follows (line 95-100):

“Lignin was purchased from NACALAI TESQUE, INC (Kyoto, Japan), and the characteristics from the company data sheet were as follows: the lignin was extracted from coniferous trees using sulfites, and chemical treatments including desulfonation, demethylation, and oxidation were performed during its production. The pH value of the product was adjusted to alkaline using sodium hydroxide. The product contains 8.0–14.0% methoxy groups and less than 15.0% water.”

Nonetheless, the purity and the molar masses were not described in the data sheet from the company.

Comment 5: The yield of the transformation must be indicated. (quantum yield, chemical yield and so on.

Answer 5: Thank you for the comment on the important aspect. Judging from total ion chromatograms analysis, the yield of a fraction containing vanillic acid can be estimated to be less than 1% of the total fraction. We added the description about the production yield to line 368-370. This low productivity must be improved in application of the laser treatment for the industrial fields in future.

Comment 6: Is the sample store in the same conditions (humidity, temperature) before irradiation and analyses.

Answer 6: Thank you for the question. The sample was stored under atmospheric conditions (about 30% humidity and 25 degree C), and the irradiation was performed under almost the same conditions in the same room. After irradiation, the sample was moved to another room for infrared microscopy, ESI-MS and NMR analyses, but these analyses were performed at room temperature and under almost the same conditions with the case of irradiation. Accordingly, it can be considered that the store conditions are not largely changed before and after the irradiation experiments.

Comment 7: It is very difficult to appreciate the changes on IR microspectroscopy observations and consequently the chemical modifications.

Answer 7: Thank you for the comment. The IR spectroscopy is traditionally used for identifying the functional groups of lignin, and many reports using the FT-IR spectroscopy for studying the structure of lignin are known (Ref. 1, 32, and 33). Nonetheless, as you suggested, it may be difficult to conclude the changes of the chemical bonds precisely by the IR spectra. The following sentence “In particular, the spectral change around 1100 cm-1 implies that C-O bond cleavage or the fragmentation of the polymerized lignin can be occurred by the irradiation” may be overstatement, and so we reduced this sentence. Instead, we changed as follows (line 182-184): “Therefore, it seems that some large conformational changes can be induced by the vibrational excitation at about 3 mm.” We think this description may be milder than the previous version.

We would appreciate it if you could consider this revision.

That’s all.

Sincerely yours

Takayasu Kawasaki, Ph.D.

Accelerator Laboratory, High Energy Accelerator Research Organization, 1-1 Oho, Tsukuba, Ibaraki 305-0801, Japan.

Phone: +81-29-864-5200-2014, Fax: +81-29-864-3182

Round 2

Reviewer 1 Report

The authors made the corrections, which greatly improvised the manuscript Degradation of Lignin by Infrared Free Electron Laser” presents results about obtaining lignin derived-phenols”.

Reviewer 2 Report

The authors have followed the recommendations.

I propose to accept it in present form.